# A Novel Risk Calculator to Predict Erectile Dysfunction in HIV-Positive Men

**DOI:** 10.3390/jpm13040679

**Published:** 2023-04-18

**Authors:** Narcis Chirca, Anca Streinu-Cercel, Marius Stefan, Justin Aurelian, Cristian Persu

**Affiliations:** 1“Carol Davila” University of Medicine and Pharmacy, Str. Vasile Lascar, Nr. 37, Sector 3, 035648 Bucharest, Romania; narcis.chirca@umfcd.ro (N.C.); cristian.persu@umfcd.ro (C.P.); 2Department of Urology, “Prof. Dr. Th. Burghele” Clinical Hospital, Sos. Panduri Nr. 20, Sector 5, 05324 Bucharest, Romania; 3National Institute of Infectious Diseases “Prof. Dr. Matei Bals”, Str. Doctor Calistrat Grozovici, Sector 1, 021105 Bucharest, Romania; 4Faculty of Applied Sciences, Polytechnic University of Bucharest, Splaiul Independentei Nr. 313, Sala BN108, Sector 6, 060042 Bucharest, Romania

**Keywords:** risk calculator, HIV infection, erectile dysfunction, sexual disorders, IIEF, QoL

## Abstract

HIV infection is considered to be a lifelong medical condition, requiring follow-up and treatment for decades. HIV-positive men are reported to have erectile dysfunction more often than age-matched healthy controls, and improving sexuality is known to potentially improve overall health-related quality of life. The aim of this paper is to evaluate the presence of ED in HIV-positive men and the associated contributing factors and to create a statistical model to assess the risk to develop ED in this population. In a prospective study, we analyzed a group of HIV-positive men in a cross-sectional manner, looking at demographics, blood test results, and smoking habits. Data were statistically analyzed using the Kruskal–Wallis test. In our series, the overall incidence of ED was 48.5%, increasing with age. Our analysis showed no correlation with blood sugar level, but a very strong correlation with total serum lipids. We were able to develop and validate a risk calculator for ED in HIV-positive men.

## 1. Introduction

Since the introduction of antiretroviral treatments, the perspective of HIV-infected patients has changed significantly, leading to an increase in disease-free period and overall survival. By controlling the viral load and the immune status, the patient’s life is no longer endangered in the short term [1,2,3,4]. Improving the immune status of these patients is associated with a decrease of the comorbidities associated with HIV, improving their survival [5,6]. 

Although for some other viral infections, some treatment options with real chances of complete cure are now available, HIV infection still remains a chronic disease that requires periodic evaluations, as patients need daily administration of antiretroviral treatment [6]. Thus, it is important that the care plan of these patients includes aspects related to establishing and maintaining a health-related quality of life as good as possible while keeping them safe [7].

Being a diagnosis that threatens the patient’s life in the long run, there is a general tendency to give less priority to the aspects related to the quality of life of these patients. The issues related to sexual activity have an important role in maintaining a good quality of life, with these patients having problems related to family, social or professional activities, and also a severe loss of self-esteem [8]. Sexuality is demonstrated to be an important aspect of the day-to-day life in all types of patients, and a good example here are the patients with spinal cord injury, where both mobility and sensations are abolished, so sexual activity might seem as being pointless; nevertheless, it is now accepted that sexuality should be stimulated and improved as much as possible in this particular population, with a significant impact on all aspects of their quality of life [7].

Many studies have demonstrated the existence of sexual dysfunction associated with HIV infection, with about 30% to 60% [8,9,10] of HIV-positive men reporting to have erectile dysfunction of some degree. Scholars could not reveal with precision the responsible mechanism, but clinical evidence shows an association with endocrine disorders, antiretroviral therapies, lipodystrophy, depression, smoking, drug use, age, and testosterone levels. The variability of the results may come from the fact that the groups of patients with differences at many levels are included in the analysis [9,11,12,13,14].

Another problem that arises is the potential low compliance with treatments due to these disturbing factors that may affect the quality of life, thus leading to poor outcomes in the long run involving all aspects of the patients’ medical condition [12,15]. Thus, clearly identifying the factors responsible for the onset of erectile dysfunction (ED) is still a challenge, with the available literature in this field lacking clear results. The tendency towards offering the patient a personalized evolution prognostic instead of general statistics pushes contemporary research towards developing mathematical models and risk calculators, using complex statistics, based on blood tests or imaging results from the patient.

The aim of our paper is to evaluate the presence of ED in HIV-positive men and the associated factors leading to it and to create a statistical model that assesses the risk to develop ED once infected with HIV, in an attempt to highlight some unresearched connections and to help clinicians predict and potentially prevent the onset of sexual disorders. Ultimately, we hope our calculator will be included in the clinical evaluation of HIV-positive males who are sexually active, and a discussion of a potential future onset of ED will be made during the initial evaluation. External validation of such a calculator will probably be required before entering standard of care evaluation.

## 2. Materials and Methods

Between 2014 and 2019, we prospectively analyzed a group of HIV-positive men, being under medical surveillance and receiving daily antiretroviral treatment. All the patients enrolled into the study were sexually active for at least one month prior to the study and older than 18 years at the moment of the informed consent signature, which was obtained from all subjects enrolled. We enrolled consecutive male patients, confirmed with HIV infection and sexually active as per the criteria described above. Patients with a previous diagnosis of ED, with or without treatment, were excluded, as per our protocol. The study is the result of interdisciplinary research and cooperation within our university, involving the departments of urology, infectious diseases, and statistics of the “Carol Davila” University of Medicine and Pharmacy in Bucharest, Romania. The statistics were overseen by a professional from the Polytechnic University of Bucharest. Prior to any study-related activity, we obtained approval from the Ethics Committee of our University.

The initial evaluation was performed in the infectious diseases department and patients were referred to the urology department for specialized evaluation for ED. Patients offered data about their medical history, including concomitant conditions and medications. We included sexually active men with no documented comorbidities or medication except antiretrovirals. Included patients filled in the International Index of Erectile Function (IIEF) questionnaire, consisting of 15 questions with standard answers with scores from 1 to 5. The erectile function is assessed by questions 1, 2, 3, 4, 5, and 15, with a maximum score of 30. The interpretation of the score is presented in Table 1, a value lower than 26 indicating the presence of some degree of erectile dysfunction, and a value from 26 to 30 being accepted as no impairment related to erectile function [16].

The IIEF questionnaire is linguistically validated, it addresses all the important components of male sexuality (erection, orgasm, desire, intercourse and overall satisfaction), and is psychometrically sound. It has demonstrated good sensitivity and specificity for detecting treatment-related changes in patients with erectile dysfunction. Very high degrees of internal consistency were observed for each of five sections, but also on the total scale, with Cronbach alpha values above 0.73 and above 0.91, respectively [17].

The evaluation protocol also included the assessment of smoking status, blood sugar level measurement, and lipidic profile evaluation. All acquired data became part of the same patient’s file and were consolidated before the final analysis.

The data obtained were statistically analyzed using the Kruskal–Wallis test, valid for a *p* value lower than 0.05 with the intention to correlate the degree of ED with as many parameters as possible while maintaining statistical significance and power. We used the IBM SPSS Statistics software platform (IBM, New York, NY, USA).

## 3. Results

Our study included a total of 103 HIV-positive men. After the initial anamnesis, where they gave information related to medical history, daily medication, and sexual activity, they completed the IIEF questionnaire assessing the erectile function (scores from questions 1, 2, 3, 4, 5, 15) according to Table 1. Related to the associated factors, we decided to analyze if factors such as age, smoking status, sugar blood level, and total lipid level have an influence on the onset of ED, and if some of them prove to be statistically valid, to use them for estimating the probability of ED installation.

The descriptive, demographic results of the included patients are described in Table 2.

After applying the IIEF questionnaire, the data regarding the erectile function are described in Table 2, and according to the categories of ED, the results are represented in Figure 1. The incidence of ED in our series is a striking 48.5%, a figure much higher than the average incidence in non-HIV, healthy males.

Regarding the age of the patients included, details are presented in Figure 2 and Figure 3 with specific values in Table 3. 

Regarding these results, we observed that men with ED tend to be older, the median age (39.86 vs. 33.24 years) being higher in the group with ED. To verify the statistical validity of this statement, we evaluated the group of HIV-positive men as binary values (ED-no or ED-yes), and using the Kruskal–Wallis test for a *p* value set at 0.05, we analyzed the data presented in Figure 2 and identified that this observation is statistically very significant (*p* = 0.0016), as presented in Table 3 Point 1.

When assessing the group as age categories, for the IIEF score (erectile function), the data show that the median IIEF score for erectile function was lower with an increasing age, as shown in Figure 3. This observation was statistically tested using the Kruskal–Wallis test for a *p* value set at 0.05 and the result is presented in Table 3 Point 2, demonstrating a *p* value of 0.0264, suggesting the significance of the statement, so again, the ED is more common with an increasing age.

We proved that both observations have statistical significance, permitting the conclusion that erectile function impairment is more frequent with aging for HIV-positive men. We speculate that this is not valid only due to the HIV presence, as a similar pattern is observed in the general population as well, but on the other hand, age is a factor that cannot be ignored.

Regarding the smoking status and serum glucose levels, the results are presented in Table 2, demonstrating that 67% (*n* = 69) of included patients were smokers and the median glycemia level was 92.23 ± 14.98 mg/dL. 

To study the correlation between smoking and erectile function, we divided the group into smokers and non-smokers (smoking variable) to study the IIEF score (erectile function) in both subgroups, with the results presented in Figure 4. For testing the statistical correlation between smoking and the IIEF score, we used the Kruskal–Wallis test for a *p* value set at 0.05 and the result, presented in Table 4 Point 1, revealed no statistical validity. 

When testing the relationship between blood glucose level and the erectile function, we decided to study the level of glycemia in the two subgroups (ED-yes and ED-no), with distribution performed with the criteria presented in Table 1, and the results are described in Figure 5. To test the statistical validity of the correlation between glycemia level and the erectile function, we used the Kruskal–Wallis test for a *p* value set at 0.05 and the result, presented in Table 4 Point 2, revealed no statistical correlation.

Regarding the serum level of total lipids, all patients provided blood samples for this evaluation, with a normal value for our laboratory ranging from 400 to 800 mg/dL and being calculated with the formula, total lipids = 2.25 × cholesterol + triglycerides + 90, and the results are presented in Figure 6 and Figure 7.

We observed that with a higher level of total lipids, the ED is more frequent, the median value for total lipids being higher in the ED—yes subgroup, and we tested if there is a statistically significant correlation between the level of total lipids and the impairment of erectile function using the Kruskal–Wallis test for a *p* value set at 0.05. 

First, the variable of erectile function was evaluated as a binary value (ED—yes and ED—no), with results presented in Table 5 Point 1, and secondary as categories (no ED, mild, mild to moderate, etc.), with the results presented in Table 5 Point 2, and in both evaluations, a strong correlation between the level of total lipids and erectile function was highlighted, with *p* values being lower than 0.05. Thus, we demonstrated that ED has a higher incidence when the level of total lipids is higher for HIV-positive men. 

Analyzing the results, it can be observed that ED is dependent on age and the level of total lipids, so using these two independent variables, we tried to develop a model to estimate the probability for a man with HIV infection to develop erectile dysfunction considering the statistical validity of the correlation between these variables and the erectile function. 

To reach this result, a logistic regression model was used, and the results are presented in Table 6. After we demonstrated and confirmed all the correlations, we were able to elaborate the mathematical formula which assesses the risk to develop ED for a HIV-positive man (Figure 8), which was the primary end point of our study.

Thus, we can estimate that when age increases by one year and total lipids remain at the same level, the probability of erectile dysfunction increases by 6.4% (exp value is 1.064)—see Table 6.

Using the model developed by us, we can estimate the probability to develop ED for all HIV-positive men included in the study, considering the age and total lipids level as shown in Figure 9, where the increase in probabilities can be observed with increasing age and the total lipids level.

## 4. Discussion

Starting from our own experience and clinical evidence from the specialized literature, confirming that men with HIV suffer from sexual disorders more often than age-matched, healthy controls [8,9,10,18], we tried to evaluate this aspect in our population and also to identify some associated factors that can influence erectile function enough to allow the design of a statistical model that can permit the evaluation of the risk of developing ED for HIV-positive men. 

Our study was able to demonstrate the correlation between the serum level of total lipids and the chances of developing ED. Age is also a known factor when it comes to evaluating the deterioration of erections. In the era of personalized medicine, our paper offers the clinician a tool able to calculate exactly the probability to develop ED for an HIV-infected male.

Analyzing the data presented in the specialized literature, there is a large number of authors who conclude the importance of age, dyslipidemia, diabetes, and smoking or drug use for erectile function impairment in the case of HIV-positive men [19,20,21,22,23,24,25]. On the other hand, it was also demonstrated that these factors can also be a cause of concern for erectile function in the case of HIV-negative men, but the results indicate that the impairment is more present when associated with HIV infection [26]. To date, we have not found any other risk calculator for ED in this particular population of patients.

Thus, taking into account all the known factors to date, we conducted our study to verify whether factors such as age, smoking, blood glucose level, and total lipid level can have a determinant influence on erectile function, allowing the creation of a statistical model for assessing the probability of developing ED.

In order to avoid secondary factors that can affect erectile function such as other comorbidities and especially those related to the presence of HIV infection, we included in the analysis only patients without other documented diseases or daily treatments other than antiretrovirals.

The preliminary results, presented in Table 2, demonstrate values in accordance with those from other studies in the literature, with ED affecting approximately one in two men in the analyzed group, most of them suffering from mild and mild to moderate ED, as shown in Figure 1, a result that is in line with those reported by other authors, where ED affects between 30–60% of HIV-positive men [8,9,10,18,27].

Regarding the related factors, we tried to test the influence of age, smoking, glycemia, and the total lipid level on erectile function. The median levels for glycemia and total lipids are within normal values, signaling that the group included in the study is in accordance with the initial plan not to include patients with other comorbidities that can influence erectile function, such as diabetes or dyslipidemia [28,29,30,31]. 

It is well known that erectile function is affected by aging, and we considered it useful to verify if our analysis group has the same tendency [32]. As presented in Figure 2 and Figure 3, as well as the statistics from Table 3, aging is a factor that influences erectile function, a result with statistical validity (*p* = 0.0016 and *p* = 0.0264). Although it was a predictable result, this verification allows the analysis of the veracity of the data obtained and, at the same time, reinforces the belief that age is a factor that should not be missing from our evaluation.

In order to evaluate the effects of smoking and glucose blood level on erectile function, the results presented in Figure 4 and Figure 5, and Table 4 demonstrate that there is no statistically valid correlation between them and ED in our group of HIV-positive men. Previous data from the literature shows that very high blood sugar levels are associated with severe ED, but no such patients were included in our study [28,31].

Regarding the association between the presence of erectile dysfunction and the level of total serum lipids, the results indicated that a higher level of the total lipids is associated with a greater impairment of the erectile function, as presented in Figure 6 and Figure 7, a statement that proved to be statistically very significant for measurements using both the binary form of erectile function (*p* = 0.0006) and the categorical form (*p* = 0.0132). These results prompt us to state with certainty that the level of total lipids is positively correlated with erectile dysfunction.

The results in the field do not accurately explain the pathophysiology, but they also associate dyslipidemia with erectile disorders. The elevated levels of cholesterol and triglycerides may be responsible for lipodystrophy and insulin resistance, which can cause ED, as previous studies have demonstrated [19]. 

After analyzing these results, we can state that in the case of men with HIV infection, age and the level of total serum lipids are statistically associated with erectile dysfunction, so we can state that some of the main risk factors for the occurrence of erectile dysfunction are an older age and an increased level of total lipids. This aspect should be discussed and proactively managed, thus offering a prevention instrument as well.

Thus, using the statistical model presented in Figure 8 for calculating the probability of erectile dysfunction, we can estimate the risk of developing erectile dysfunction depending on the level of total serum lipids and the age of the patient. Adapting this result to clinical practice, we consider it may be a useful tool for medical doctors, who will be able to discuss the risk of developing ED in a personalized way and in an attempt to reduce the risk of erectile dysfunction by lowering total lipid levels.

Some limits of this paper were identified and analyzed. First, the relatively low number of the patients included, as well as some age categories including only a small number of men. Second, the results of blood tests in our series, even when they were abnormal, were within reasonable ranges, so extreme values might compromise the effectiveness of our risk calculator. Another limit we identified is not directly related to our work, being more of a typical issue when dealing with ED—the different perception of the patient towards his symptoms and a significant intra-individual variability, making it possible to report very different perspectives of the same situation at different time points. Psychogenic ED affects a significant percentage of the general population and this percentage is probably similar or higher in HIV-infected males. Since no objective evaluation can be carried out in this aspect, we speculate that our statistical analysis remains valid for organic ED and temporary episodes of psychogenic ED might come and go unpredicted.

## 5. Conclusions

ED affects a significant percent of all HIV-positive men, with approximately one in two men suffering from some degree of impairment of their erectile function. The main risk factors that could lead to this impairment are age and total serum lipid level, which demonstrated statistically significant correlations in our study.

Ensuring a good quality of life for HIV patients is one of the goals of their medical care and preventing or correcting erectile dysfunction must be part of this plan, contributing to the overall compliance to any treatment. The statistical model we developed allows an establishment of the risk for developing erectile dysfunction and offers a base for preventive measures as well. Future studies on larger populations of patients are needed to further validate the formula we propose, but external validation is definitely required before it enters daily practice.

## Figures and Tables

**Figure 1 jpm-13-00679-f001:**
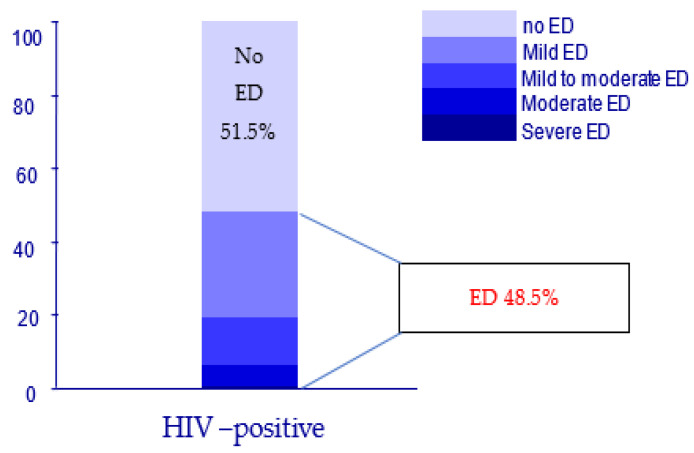
ED distribution by IIEF score categories in the IV-positive group.

**Figure 2 jpm-13-00679-f002:**
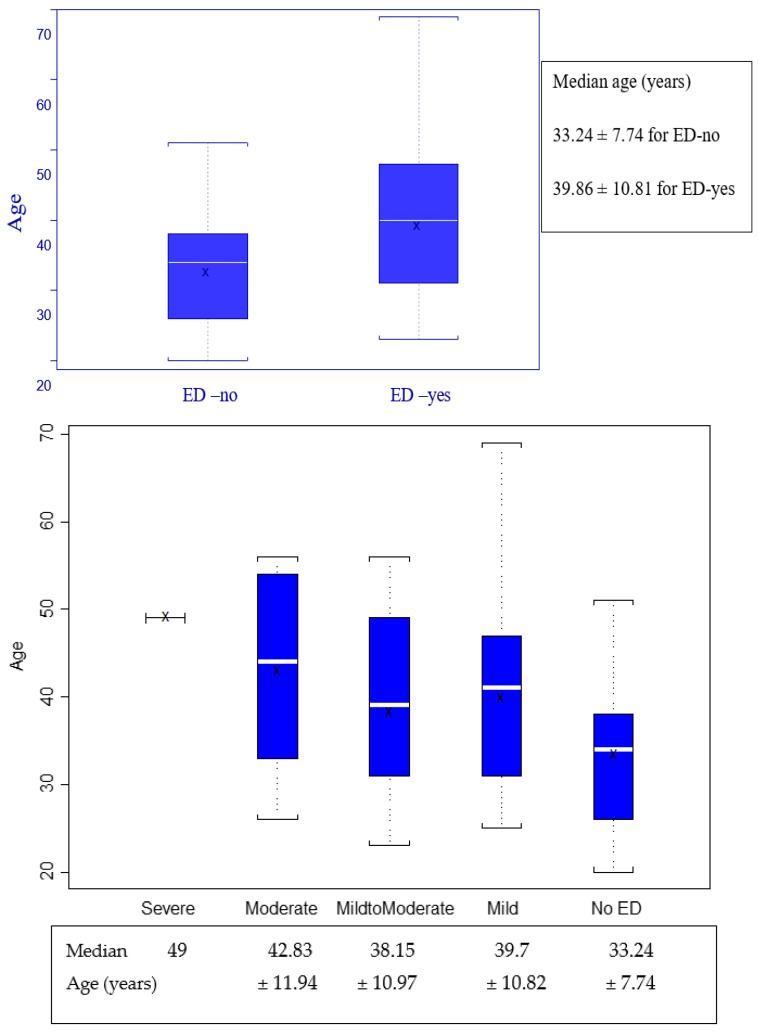
Age distribution related to erectile function.

**Figure 3 jpm-13-00679-f003:**
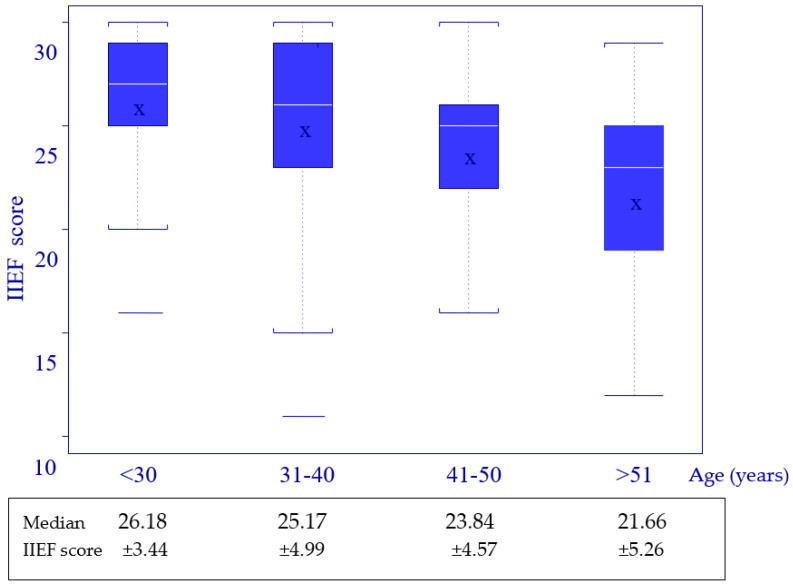
IIEF score (erectile function) distribution by age groups.

**Figure 4 jpm-13-00679-f004:**
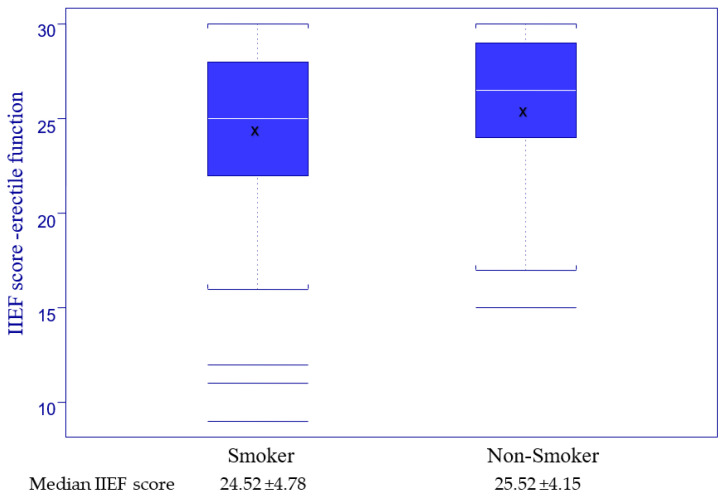
IIEF score (erectile function) distribution related to smoking.

**Figure 5 jpm-13-00679-f005:**
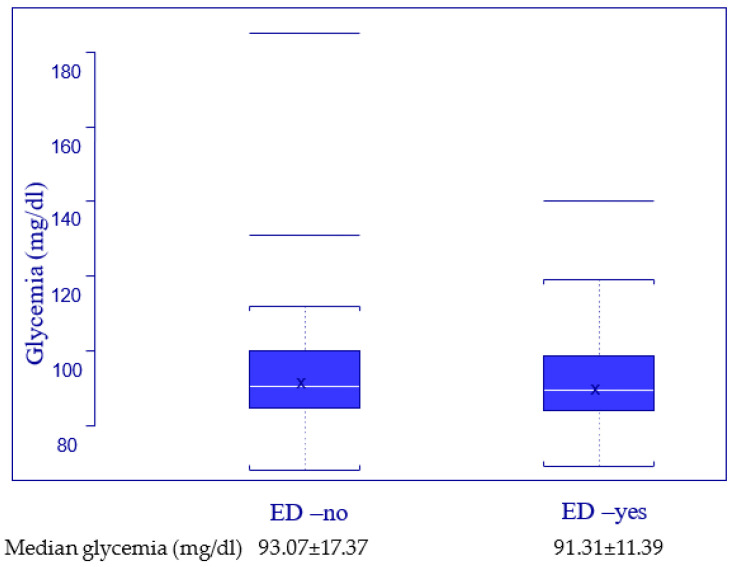
Glycemia related to erectile function.

**Figure 6 jpm-13-00679-f006:**
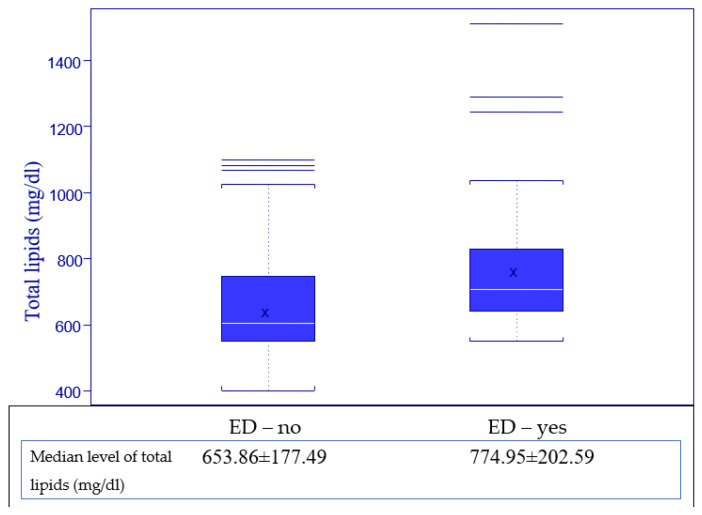
Distribution of total lipids related to erectile function.

**Figure 7 jpm-13-00679-f007:**
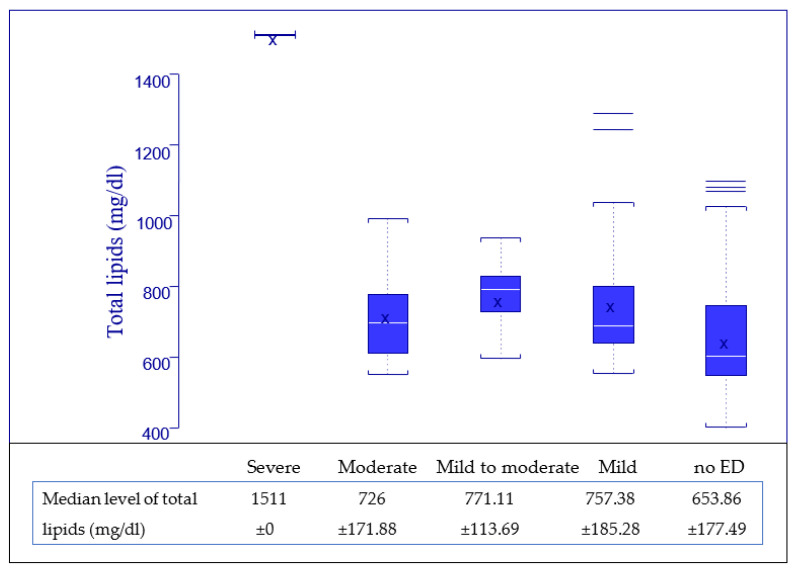
Distribution of total lipids related to erectile function categories (Table 1).

**Figure 8 jpm-13-00679-f008:**
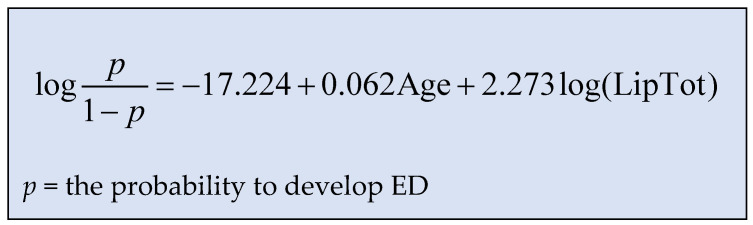
The probability model for assessing the ED.

**Figure 9 jpm-13-00679-f009:**
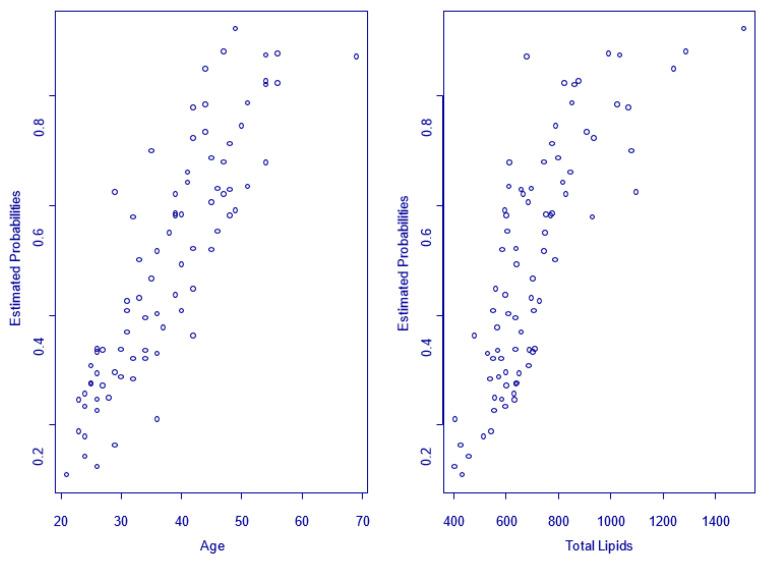
The probabilities to develop ED for all patients included related to age and total lipids.

**Table 1 jpm-13-00679-t001:** Interpretation of IIEF—Erectile Function (questions 1 to 5, 15—processed after IIEF interpretation of Rosen 1997 and Cappelleri 1999).

Score	Interpretation
26–30	no ED
22–25	mild ED
17–21	mild to moderate ED
11–16	moderate ED
6–10	severe ED

**Table 2 jpm-13-00679-t002:** Characteristics of the HIV-positive men.

Parameter	Patients/Median Value
Total patients	103
Age (median)	36.45 ± 9.88 years
BMI (median)	23.7 ± 4.1 kg/m^2^
Glycemia (median)	92.23 ± 14.98 mg/dL
Total lipids (median)	712.96 ± 198.56 mg/dL
Smoker	69 (67%)
Non Smoker	34 (33%)
No ED	53
With ED	50
ED prevalence	48.5%

**Table 3 jpm-13-00679-t003:** Correlation between age and ED.

Kruskal–Wallis Chi-Squared		DF	*p*-Value
1	9.9434	1	*p* = 0.0016
	>mean(Age & ED == “ED yes”]) : [1] 39.86 >mean(Age & ED == “ED no”]) : [1] 33.2452830188679		
2	9.2325	3	*p* = 0.0264
	>mean(IIEFscore & Age-Factor == “<30”]) : [1] 26.1818181818182 >mean(IIEFscore & Age-Factor == “31-40”]) : [1] 25.1714285714286 >mean(IIEFscore & Age-Factor == “41-50”]) : [1] 23.8461538461538 >mean(IIEFscore & Age-Factor == “>51”]) : [1] 21.6666666666667		

**Table 4 jpm-13-00679-t004:** Correlation between smoking/blood glucose and ED.

	Kruskal-Wallis Chi-Squared	DF	*p*-Value
1. Smoking	1.0233	1	*p* = 0.3117
>mean(IIEFscore & Smoking == “Yes”]) : [1] 24.5217391304348 >mean(IIEFscore & Smoking == “No”]) : [1] 25.5294117647059			
2. Glycemia	0.2682	1	*p* = 0.6046
>mean(Glycemia & ED == “ED-yes”]) : [1] 91.3125 >mean(Glycemia & ED == “NoED”]) : [1] 93.0769230769231			

**Table 5 jpm-13-00679-t005:** Correlation between the level of total lipids and ED.

Erectile Function	Kruskal-Wallis Chi-Squared	DF	*p*-Value
Binary (yes, no)	11.7778	1	*p* = 0.0006
>mean(TotLip & ED == “ED-yes”]) : [1] 774.951219512195 >mean(TotLip & ED == “ED-no”]) : [1] 653.860465116279			
Categorial	10.7398	1	*p* = 0.0132
>mean(TotLip & ED-Cat == “Severe”]) : [1] 1511 >mean(TotLip & ED-Cat == “Moderate”]) : [1] 726 >mean(TotLip & ED-Cat == “MildtoModerate”]) : [1] 71.111111111111 >mean(TotLip & ED-Cat == “Mild”]) : [1] 757.384615384615 >mean(TotLip & ED-Cat == “NoED”]) : [1] 653.860465116279			

**Table 6 jpm-13-00679-t006:** Logistic regression: statistical results.

	Intercept	Age	Log (Total Lipids)
Coefficients Value	−17.2248	0.0625	2.2736
Std Error	7.2814	0.0279	1.1580
t-value	−2.3655	2.2339	1.9633
*p*-value for Wald tests (significance test)	0.0180	0.0254	0.0496
>exp coeff	3.3062	1.0645	9.7143

## Data Availability

The data presented in this study are available on request from the corresponding author. The data are not publicly available due to privacy reasons and local regulations.

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
