# Peer review of "A Novel Risk Calculator to Predict Erectile Dysfunction in HIV-Positive Men"

_jpm, 2023, doi:10.3390/jpm13040679_

Round 1
Author Response
Please find our reply in the word document below.

Reviewer 2 Report
This is an observational prospective study. There are ethical and methodological major concerns.
Ethical flaws
1) There is no study registration or institutional review board approval number
Methodological flaws
1) The enrolment modality has not been described
2) A primary outcome has not been defined neither a population size was fixed to assess a pre-defined effect on the outcome
Therefore, even if it is an observational study, it should be considered as retrospective analysis of case series.
That said, the topic is interesting as well as the findings that may be used as a guide for future studies. Finally, an external validation for the statistical model is needed to support any conclusion about its applicability
Author Response
Please find our reply in the attached file.

Round 2
Reviewer 1 Report
The authors revised the manuscript very carefully and took all of my recommendations into consideration. From my point of view, the paper is of good quality and could be accepted for publication
Author Response
Thank you very much for taking your time to analyze our abstract and for your kind resolution.
Reviewer 2 Report
Authors amended the paper properly. It is now clear their primary objective. It should be stressed in the methods and in the conclusion sections that an external validation is needed before to confirm any clinical application.
Author Response
Thank you for your very kind reply.
We updated the M&M and Conclusions as per your suggestion.
Thank you again!